# Slice-100K: A Multimodal Dataset for Extrusion-based 3D Printing

**Anushrut Jignasu**[1][†]    **Kelly O. Marshall**[2][†]    **Ankush Kumar Mishra**[1]
**Lucas Nerone Rillo**[1]    **Baskar Ganapathysubramanian**[1]    **Aditya Balu**[1]
**Chinmay Hegde**[2][*]    **Adarsh Krishnamurthy**[1][*]

{ajignasu | akmishra | lucasnr | baskarg | baditya | adarsh}@iastate.edu
{km3888 | chinmay.h}@nyu.edu

[1]Iowa State University    [2]New York University

[†]Equal contribution    [*]Corresponding authors

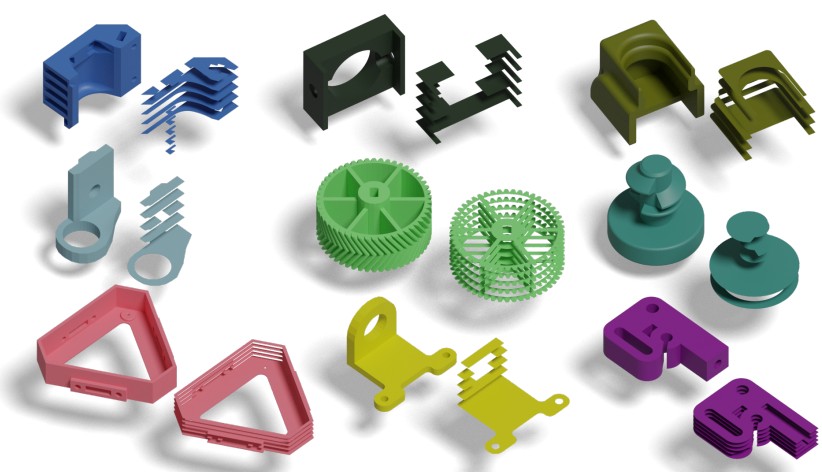

**Figure 1:** *The Slice-100K dataset consists of STL files and their G-code counterparts. Each pair here consists of STL (left) and its slices (right) for G-code.*

## Abstract

G-code (Geometric code) or RS-274 is the most widely used computer numerical control (CNC) and 3D printing programming language. G-code provides machine instructions for the movement of the 3D printer, especially for the nozzle, stage, and extrusion of material for extrusion-based additive manufacturing. Currently, there does not exist a large repository of curated CAD models along with their corresponding G-code files for additive manufacturing. To address this issue, we present Slice-100K, a first-of-its-kind dataset of over 100,000 G-code files, along with their tessellated CAD model, LVIS (Large Vocabulary Instance Segmentation) categories, geometric properties, and renderings. We build our dataset from triangulated meshes derived from Objaverse-XL and Thingi10K datasets. We demonstrate the utility of this dataset by finetuning GPT-2 on a subset of the dataset for G-code translation from a legacy G-code format (Sailfish) to a more modern, widely used format (Marlin). Our dataset can be found here. Slice-100K will be the first step in developing a multimodal foundation model for digital manufacturing.

38th Conference on Neural Information Processing Systems (NeurIPS 2024) Track on Datasets and Benchmarks.

# 1 Introduction

In recent years, the integration of digital design and computer-aided manufacturing processes has led to groundbreaking innovations in the manufacturing sector [1, 2]. One of the most transformative technologies at this intersection is additive manufacturing or 3D printing, which enables the physical manufacturing of digital assets [3, 4]. 3D printing surpasses the limitations of traditional manufacturing techniques by enabling the creation of parts with complex geometric shapes [5, 6]. A commonly used 3D printing method is extrusion-based additive manufacturing [7, 8], often based on Fused Deposition Modeling (FDM) for manufacturing plastic or polymer parts. In this method, bits of thermoplastic material are sequentially extruded from a heated nozzle, which has three degrees of freedom. The nozzle moves in a flat 2D plane and builds up the desired shape layer-by-layer.

A typical 3D printing process begins with creating a 3D model of the part in a computer-aided design (CAD) program. This CAD model is then usually exported as a triangulated mesh file (for example, STL, PLY, or OBJ). The triangulated model is then "sliced" into multiple layers based on the resolution or the layer height of the 3D printer. Each layer is then converted into a sequence of programmatic instructions for the movement of the 3D printer's nozzle and extrusion of material along the boundary or "contour" of each layer. The instructions also include the movement of the nozzle and extrusion of material inside the contours or the "infill." These instructions are then directly sent to the 3D printer for physical manufacturing. The most common representation for storing this information is **G-code** (Geometric code) or RS-274, a computer numerical control (CNC) programming language. G-code provides machine instructions for the movement of the 3D printer, especially for the nozzle, stage, and extrusion of material for extrusion-based additive manufacturing. Although some extensions of G-code have been written to include basic abstractions such as for-loops, the vast majority of G-code in use consists mainly of low-level instructions that provide a sequence of commands to be carried out by the 3D printer.

Since 3D printing is a layered manufacturing process, it requires performing the slicing process. The slicing process operates on the entire object and splits it along the print direction (usually the Z-axis by default). Each layer is then used to generate the printer instructions for contour and infill. However, achieving high-quality fabricated models often requires manual tuning of the slicing software. The iterative improvement of a given G-code file to produce a 3D-printed model that exactly matches its CAD representation is a non-trivial challenge. In addition, there are several "flavors" of G-code files depending on the compatibility of the 3D printer's controller hardware. Due to the low-level nature of G-code, manually debugging a G-code file is cumbersome, if not impossible. Features such as line-level and layer-level natural language comments are infrequent. While custom solutions such as regular expression matching could be leveraged for correcting G-code, they fall under a rigid set of methods and are not generalizable.

In the last few years, while advances in AI have impacted various domains, their potential in computer-aided design (CAD) and cybermanufacturing remains largely untapped. Modern LLMs and Vision-Language Models (VLMs) could provide an avenue to realize this potential. The ability of LLMs to process, comprehend, and generate natural language descriptions, code, and other text data can be leveraged to interpret, generate, and manipulate G-code. LLMs for 3D shape modeling have been shown to enable operations on meshes [9, 10] and point clouds [11, 12]. G-code, with its unique language-based structure, presents distinct challenges for machine learning, mainly due to the context window limitations of current LLMs. Many existing deep-learning-based computer vision applications leverage 2D datasets (images), text descriptions, or a combination of such modalities for both supervised or self-supervised pre-training of foundation models (see Table 1). However, none of these datasets provide a curated avenue for training a manufacturing domain-specific foundation model.

To bridge this gap, we introduce Slice-100K, a curated multimodal dataset (see Figure 2 for reference) of G-code, CAD models, renderings, and geometric properties to facilitate the application of VLMs for additive manufacturing. We believe Slice-100K will encourage the research community to address new problems in the design and manufacturing space. Our dataset, built using models from Objaverse-XL and the Thingi10k dataset, encompasses a diverse range of 3D printable objects and provides a comprehensive resource for training a manufacturing domain-specific foundation model.

**Contributions:** This paper introduces Slice-100K, a multimodal dataset for manufacturing applications. The main features include a first-of-its-kind curated dataset of more than 100,000 G-code files along with their corresponding STL CAD files, renderings, LVIS (Large Vocabulary Instance

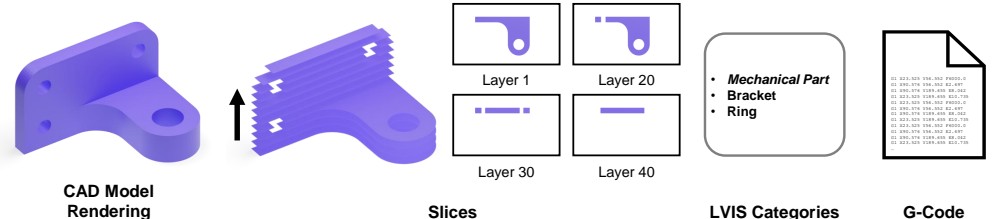

**Figure 2:** *Different data formats in Slice-100K. We build our dataset using CAD models (STL files) and their renderings. Furthermore, we slice these STL files to generate G-code (build direction shown by black arrow) and their categorical classifications.*

Segmentation) categories, and geometric properties. We demonstrate the utility of Slice-100K by evaluating existing LLMs for G-code geometric transformations. We also showcase a novel application of our dataset on LLM-based G-code flavor translation. We believe that this multimodal dataset will be the starting point for a foundation model in digital manufacturing.

## 2   Background and Related Work

**G-code:** G-code forms a crucial intermediary between digital design and physical manufacturing, providing an expressive language-based representation for 3D objects. For example, the most straightforward G-code command is `G1`, which directs the 3D printer to move its nozzle towards a spatial coordinate. This is usually followed by a coordinate in the form `Xaaa Yaaa`, where movement along the **X** and **Y** axes are given by a specific numeric value `aaa`. For extrusion-based 3D printers, a thermoplastic material is extruded from a heated nozzle that has three degrees of freedom. An example extrusion move is given by `G1 X50.6 Y36.2 E2.3`, where the nozzle moves 50.6 units along **X**, 36.2 units along **Y** and extrudes 2.3 units of material. Other commands instruct the printer to change settings, such as the material/ink feed rate, or perform more complex movements without extruding material.

**Language Models for Code:** LLMs have also been used for programming language analysis and code generation. Coding-focused LLMs are mainly trained on a mix of web-scraped data, coding repositories, and instructions and often surpass general-purpose LLMs in code-related tasks. Current research has lead to many such models [13–18]. Most notable ones include WizardCoder [18], Code Llama [19], and Instruct-CodeGen [16]. Codex [20] is an early model deployed under Github's Copilot feature and acts as an IDE assistant that can understand local code context, make suggestions, and generate entire blocks of code.

**3D Datasets:** The current research community has proposed and leveraged various 3D datasets [21–28]. Notable ones include Objaverse 1.0 [23] and Objaverse-XL [24], with the former consisting of over 800K 3D models with higher quality textures and geometry types (Table 1). The latter is a massive dataset of over 10 million objects gathered from various sources, including Thingi10K and GitHub repositories. The diversity of objects in terms of shapes and categories is an advantage for Objaverse-XL. Most of the datasets currently used by the research community provide a single modality (meshes or voxels), and some include text descriptions and renderings for visual supervision tasks. However, none of the currently available datasets provide curated assets for encouraging research in the manufacturing domain. The largest public G-code dataset we are aware of is the Greater G-code [29] dataset, which only contains 860 G-code files paired with their STL renderings.

**LLMs and 3D Datasets:** Language understanding methods have been applied in the 3D domain for a wide array of tasks including 3D captioning [26, 30], object grounding [11, 31], 3D conversation [32], and text-conditioned generation [9, 10, 10]. Recently, there has been a surge of interest in multimodal large language models (MLLMs). MLLMs combine the language-based reasoning and knowledge of LLMs with the ability to comprehend other data modalities. Vision-augmented LLMs [33–35] encode images into an LLM's embedding space. These methods have been subsequently extended to the 3D domain for different forms of 3D representation, such as point clouds [12, 36], and sparse outdoor LiDAR data [37]. Paschalidou et al. [38] use a transformer-based model (not LLM) to predict 3D objects in a scene autoregressively. 3DLLM [11] maps 3D scenes to a set of 2D image embeddings and uses a query-token embedding technique based on BLIP-2's Q-Former [34] to perform a diverse set of 3D-related tasks. GPT4Point [36] also leverages a similar Q-Former for point-

**Table 1:** *Comparison of different 3D multimodal datasets currently available.*

| Dataset | Mesh | Renderings | Categories | G-code |
|---|:---:|:---:|:---:|:---:|
| ABC | ✓ | ✓ | ✗ | ✗ |
| ShapeNet | ✓ | ✓ | ✓ | ✗ |
| Thingi10K | ✓ | ✓ | ✗ | ✗ |
| Objaverse 1.0 | ✓ | ✓ | ✓ | ✗ |
| Objaverse-XL | ✓ | ✓ | ✓ | ✗ |
| **Slice-100K** | ✓ | ✓ | ✓ | ✓ |

text feature alignment. Chat3D [32] uses an object-centric 3D representation to train a 3D-LLM for dialogue. Feng et al. [39] does in-context learning on room layouts from the 3D-FRONT dataset [40]. PointBERT [41] did some early work on point-cloud representation learning with transformers. Fu et al. [30] align visual features from 3D scenes with text to finetune a LLaMa-2-chat-70B [42] model for scene understanding and question answering.

**LLMs for Design and Manufacturing:** Recent research has shown that natural language descriptions can be used for various tasks related to 3D printing, such as generating novel shapes [43–46], editing scenes [47], and reasoning about geometry in the volume space [48]. Makatura et al. [49] thoroughly examine GPT-4's suitability for automated design and manufacturing. Badini et al. [50] use ChatGPT to modify G-code, but they only alter the parameters in the G-code header. These modifications allow them to address common errors in the 3D printing process, such as warping, bed detachment, and stringing. Kulits et al. [51] train an LLM to autoregressively generate structured representations of simple 3D objects from the CLEVR dataset [52].

## 3 The Slice-100K Dataset

### 3.1 Data Collection Process

**Dataset:** We build our Slice-100K dataset using Objaverse-XL's openly available 3D dataset and Thingi10K dataset. Specifically, we download STL models from the Thingiverse branch of Objaverse-XL since these are solid models specifically designed to be additively manufacturable. We filter our models from the Thingi10K dataset using the following keywords: `num components = 1`, `is manifold`, and `is oriented`. A summary of our dataset is shown in Table 2, and we describe each data source below. In addition to providing STL models, our dataset includes renderings, descriptive captions, and detailed geometric properties. The metadata for each model is generated using Open3D, a library that facilitates the processing and analysis of 3D data. Key geometric properties such as vertex manifold, edge manifold, and vertex count are calculated and included in the dataset. These properties are essential for understanding the structural characteristics of the models and can be leveraged in various applications, such as model optimization and error detection in 3D printing.

The **Objaverse-XL** dataset comprises of 3D objects gathered from Github, Thingiverse, Smithsonian Institution, Polycam, and Sketchfab. We collect data from the Thingiverse subset of Objaverse-XL. Thingiverse is one of the largest online platforms consisting of user-generated digital designs and is particularly focused on 3D printable files, encouraging community interaction and collaboration. A majority of these files are provided in the STL format and are available under Creative Commons licenses. The models on Thingiverse cover a wide range of categories, including functional parts, artistic creations, and educational tools. This extensive and diverse collection makes it an invaluable resource for creating comprehensive datasets for additive manufacturing.

The **Thingi10K** dataset [27] is a collection of 10,000 3D models sourced from Thingiverse. It is curated explicitly for research purposes and provides a diverse set of models that are manifold and oriented, making them ideal for various computational geometry and 3D printing research applications. The dataset includes metadata and annotations that facilitate the development of machine learning models and other computational tools.

**Table 2:** *Composition of Slice-100K.*

| Source | Number of Objects |
|---|---|
| Objaverse-XL (Thingiverse) | 96,479 |
| Thingi10k | 3,589 |
| **Total** | 100,068 |

**G-code Generation:** The G-code generation process is a critical component of the Slice-100K dataset. We utilize PrusaSlicer's [53] command line functionality to slice all our models. Each model is sliced using two distinct G-code flavors—Sailfish and Marlin. Prusa's slicer is an open-source and widely-used slicing software that prepares 3D models for printing by converting them into G-code, which provides specific instructions for 3D printers. Additionally, it allows for extensive configuration options, allowing for fine-tuning of print settings such as layer height, infill density, and support structures. This flexibility ensures that the generated G-code is high quality and suitable for different 3D printers and printing conditions. Furthermore, to minimize our data footprint, we generate G-code files in the binary G-code (`.bgcode`) format, a functionality recently incorporated by Prusa's slicer. An important aspect of the slicing pipeline is the infill pattern selection, primarily due to its impact on total print time and structural properties of manufactured models. To encourage diversity among our G-code files with respect to structural properties, while slicing each STL file, we randomly select from four different infill patterns: (1) *Gyroid*: Empirically known to give equal strength across all directions and optimizes for a quicker print time; (2) *Honeycomb*: Uses a grid of hexagons, providing increased mechanical resistance, and non-crossing paths; (3) *Cubic*: Introduces crossing paths, potentially generating air-pockets; and (4) *Grid*: Uses a two-way checkerboard-like pattern for faster infill.

**STL Renderings:** To generate renderings of our STL files, we utilized Blender [54] rendering scripts made available by Objaverse-XL. We modified the scripts to generate a total of 10 views for each object - 6 orthogonal views (front, back, top, bottom, left, right) and 4 isometric views (captured from top four corners of a cube). Each object was rendered with a random color. These renderings were utilized for object category generation.

**LVIS Object Categories:** To generate the text category of each model in Slice-100K, we use the framework shown in Figure 3. For each model in our dataset, we assign the top 3 of the 1200+ LVIS (Large Vocabulary Instance Segmentation) categories [55]. This process helps enhance the utility of the dataset, enabling better categorization and facilitating more effective use in various research and development applications. For each CAD model, we generate multiple views using Blender. This step ensures comprehensive visual coverage of the model, capturing its geometry from various angles. The generated renderings serve as input to a pre-trained Vision-Language model to generate image embeddings. Specifically, we utilize a pre-trained CLIP-ViT-L-14 [56, 57] to obtain embeddings for each view. To integrate information from multiple views, we compute an average embedding for each object. This average embedding combines the features from all views into a single, unified representation, providing a comprehensive summary of the visual characteristics of the object.

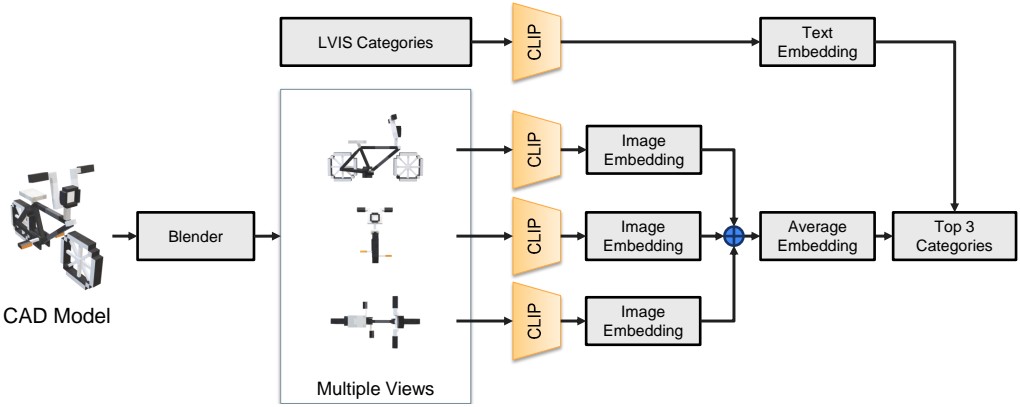

**Figure 3:** *Framework to generate the LVIS categories of the 3D objects.*

In parallel, we also process the 1200+ LVIS categories to obtain the text embeddings for all categories. Using the average embedding, we then match each object to the closest categories in the text embedding. By comparing the average embeddings, we identify the top 3 most relevant LVIS categories for each object in our dataset.

## 3.2 G-code Translation

G-code translation involves converting a G-code from one flavor to another while preserving the necessary context associated with each flavor and finding a correspondence between any two given flavors of G-code. We begin with G-code data in two different flavors, Sailfish and Marlin. Sailfish is a legacy G-code format that is not currently used by the 3D printing community. Marlin is a modern G-code format that has been heavily adopted, and in some cases, other G-code flavors are built on top of Marlin. Given this, we leverage our proposed dataset to finetune GPT-2 for the task of G-code translation from Sailfish to Marlin. G-code is inherently a low-level language, and for a task like translation, the quality of data being fed into an LLM has a significant impact on its performance. Keeping this in mind, we perform some data pre-processing to effectively maintain the context across lines of G-code.

### 3.2.1 Data Pre-Processing Methods

A major challenge in applying language-modeling-based techniques to G-code is the length of G-code files. While the G-code representation of a 3D shape can be separated into layers (which do not share information and hence be handled independently), this is still not sufficient, as a single layer can be over the token limit. This motivates methods for further splitting of G-code layers, allowing us to decompose mappings between G-code files into a series of mappings between smaller G-code portions. Crucially, these methods can be applied to different G-codes regardless of the variants they are written in while ensuring that the resulting pairs of G-code segments represent the same spatial semantics. To accomplish this, we first permute the contours in each G-code layer so that they have the same ordering. We then adaptively select portions to create matching pairs.

**Contour Flipping:** Let $L_A$ and $L_B$ be two G-code layers that use different flavors to represent the same 3D information. We can decompose each of these layers into a series of $N$ contours $c_1^{(A)}, .., c_N^{(A)}$ and $c_1^{(B)}, .., c_N^{(B)}$, each represented using their respective flavor. Both sequences contain the same set of unique contours irrespective of the flavors. Because of this, we can define a bijective mapping $M : [N] \to [N]$ such that the contour $c_{M(i)}^{(B)}$ is equivalent to $c_i^{(A)}$.

The primary challenge in our preprocessing is to find this bijection, which, once found, allows us to re-order the contours of $L_B$ so that $\forall i \in [N]$ we have that $c_i^{(B)}$ is equivalent to $c_i^{(B)}$. To determine $M$, we iterate over each contour $c_i^{(A)}$ in $L_A$ and find its corresponding contour $c_{M(i)}^{(B)}$ in $L_B$. We consider two contours to be matching if there are specific commands which are included in both. More specifically, we define a method for representing a single line of G-code so that identical representations will indicate matching contours.

To minimize the possibility of a duplicate representation (that could lead to a false match), we base this criterion on G-code lines, which contain commands to extrude at specified coordinates. Other commands are disregarded as they are likely to be repeated throughout a file or contain syntax that differs across flavors. In contrast, extrusion locations are specified using floating point coordinates with several digits of precision, making it rare for the same point to appear in different contours. We further account for the possibility of duplicate locations by concatenating the line's coordinates with the next two lines in the contour where possible. If these following lines do not contain a location-specific extrusion command, we simply include in their place a token denoting an empty line. Together, this creates a string representation of each line that strips away flavor-specific syntax while including enough contextual information to prevent unwanted duplicates.

Using this consistent characterization of G-code lines allows us to match contours by simply finding a single pair of lines with the same representation. However, due to the length of G-code layers, it is highly inefficient to consider all possible pairs of lines when looking to match contours. To alleviate this, we pre-compute a lookup table for $L_B$. For each line of a contour $c_B^{(i)}$, the lookup table maps from the line representation to the index $i$. Then, when iterating over the contours of $L_A$, we compute the representation for each line and search the lookup table. If there is a match, then we add these indices to our bijection $M$. While this contour flipping method cannot be guaranteed to

always find the correct bijection $M$ due to variations amongst some contours, we find that it is highly reliable, producing aligned G-code for over $99.9\%$ of the G-code layers in our dataset. We include pseudocode for our method in the Appendix (Algorithm 1).

**Pair Creation:** Given two G-code layers that have undergone contour flipping so that they have the same high-level semantic ordering, we can reasonably expect to divide them each into pairs of contiguous sections sharing the same 3D information. Because there are often commands included in one flavor but not the other, we cannot simply select portions of equal length and expect them to be translatable. Instead, we have to adaptively determine the cutoff points for each section.

Here we represent the layers as sequences of lines, with $L_A = \ell_1^{(A)}, ..., \ell_N^{(A)}$ and $L_B = \ell_1^{(B)}, ..., \ell_N^{(B)}$. Our goal of separating these layers into $K$ matching chunks then amounts to finding pairs of delimiting line indices $(k_i^A, k_i^B)_{i=1}^K$ so that the resulting G-code segments $\ell_{k_i^A}^{(A)}, .., \ell_{k_{i+1}^A - 1}^{(A)}$ and $\ell_{k_i^B}^{(B)}, .., \ell_{k_{i+1}^B - 1}^{(B)}$ meet our requirements. In particular, we can ensure that the segments contain all the same content as long as the beginning and end lines of each language correspond to the same commands.

Our pair creation approach finds these matching line indices while respecting a maximum length parameter (see Algorithm 2 in Appendix). In short, we iteratively find index $k_{i+1}^A$ by starting with a candidate value which is $k_{i+1}^A$ plus the maximum length. We then try to find a matching line in $L_B$ and, if successful, consider this a pair. If we cannot find a matching line for our candidate, we decrease the candidate line index by one and continue trying. We use a line representation similar to the one used for contour flipping to determine whether a pair of lines is matching.

**Handling Extrusion Values:** Through the previously described preprocessing methods we have been able to create pairs of G-code chunks which represent the same local information and can therefore be translated between. However, there is an additional non-local dependence that must be accounted for in the G-code extrusion values. In addition to telling the 3D printer where to move, a line of G-code also tells it how much material to extrude during this movement.

This is specified through an "E" command which states how much total material will have been extruded once that point is reached. For instance, if one line contains an E value of 3 and the next line has an E value of 3.1, then .1 units of material should be extruded during this movement. There are also specialized language-specific commands throughout a shape's G-code, which reset the total extrusion values to some smaller constant.

Because these values represent a cumulative sum of all material extruded up to that point starting from the most recent reset value, there is a non-locality element that must be addressed. During preprocessing, we amend each extrusion value by subtracting the previous line's extrusion value. We call this new value the relative extrusion. This represents only the amount of material that is to be extruded during this movement and allows for any translation model to learn a simple local mapping that is not dependent on other chunks. Finally, after generating the G-code in this relative form, we convert it back to its original format by computing its cumulative sum.

### 3.3 Geometric Transformation - Scaling

Scaling is considered to be a simple geometric transformation that results in a geometry being enlarged or shrunk depending on a scaling factor. We assume uniform scaling along all three principle directions (X, Y, and Z). We evaluate the ability of current chat-based LLMs to perform this simple linear transformation by providing them with a single layer of G-code and asking the prompts:

```
Can you scale the coordinates by a factor of 2 and give me the updated G-code?
Can you scale the entire layer by a factor of 2 and return the updated G-code?
```

At the time of our evaluation, we empirically arrived at the maximum number of lines of G-code an LLM in our test suite can accept before crossing their respective token limits. We leverage this fact to chunk the G-code before feeding it to an LLM.

### 3.4 Evaluation Metrics

To measure the quality of G-code generation models, we introduce an image-space IoU metric for G-code fidelity in comparison to ground truth. Because small errors in the produced G-code can lead to significant divergence in the produced shape, we find it insufficient to use a language-based metric for evaluation. Instead, we use an image-based measure of fidelity by rendering top-down images of each layer.

**G-code Renderer:** To the best of our knowledge, there does not currently exist open-source software that can programmatically generate renderings of G-code objects. To remedy this, we introduce and make public our Python-based tool for this purpose. The renderer generates a top-down rendering of an individual layer. This layer-wise approach is sensible for examining the 3D structure as it avoids occlusions and captures all relevant parts of the shape, even the infill that provides internal structural support for the part and may not be visible from the outside.

**Image-Space IoU:** We make use of our top-down renderer by defining an Intersection over Union (IoU) metric to capture similarity in image space rather than text space. To compute this loss for a layer of translated G-code, we render the layer as well as its ground-truth counterpart into top-down images and compute the 2D IoU. We can use this metric to quantify how well a G-code generation model produces accurate instructions for printing in the physical space. To account for the 3D-printing process's varying levels of sensitivity to error, we further define the IOU@$k$ metric as the percentage of translated layers that have an IOU greater than $k$.

## 4  Experiments

We use Slice-100K for two tasks: evaluating current LLMs for G-code geometric transformation (*scaling*) and G-code flavor translation by finetuning GPT-2,

### 4.1  Evaluating Existing LLMs for G-code Geometric Transformations

We evaluate some of the existing chat-based LLMs (GPT series [58, 59], Claude [60], Llama-2 [42], and Starcoder [61]) for performing geometric transformations, specifically scaling a layer of the model. We find GPT-3.5 and GPT-4 struggle with the S-shape. Claude-2 is able to generate the outer contour of the cylinder and cube but struggles with infill generation for the cylinder and the S-shape. Furthermore, we see that the open-source models—Llama-2-70b and Starcoder—do not perform well. We visualize the G-code outputs from the various LLMs in our test suite and render them using Ultimaker's Cura [62] slicing software. Our results are shown in Figure 4.

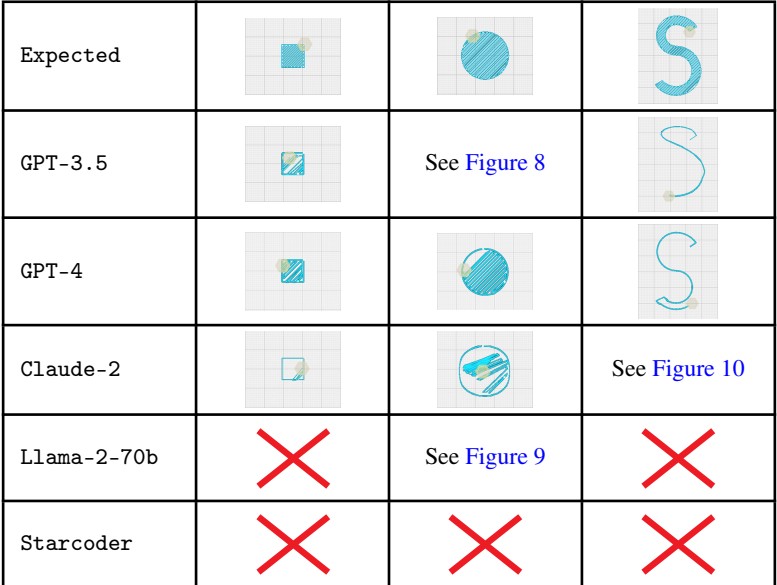

**Figure 4:** *G-code visualization for scaling operation on all LLMs. Expected G-code is shown in the top row. Please see the referenced figures in the Appendix for additional renderings.*

### 4.2  G-code Flavor Translation

For finetuning, we create a paired dataset of G-code chunks in each flavor using the preprocessing methods outlined in Section 3.2.1 using a maximum chunk size of 20 lines. We then finetune a lightweight GPT-2 model for translation using a next-token prediction loss. During inference, we do not have access to the ground-truth Marlin G-code, which would be needed to determine the cutoff lines for pair creation, so we instead split our Sailfish input into smaller chunks of fixed-length.

**Table 3:** *Performance of GPT-2 models finetuned for G-Code translation using differently sized subsets of SLICE-100k compared using IOU-based metrics. Training data reports the amount of Slice-100K data that models were finetuned on. The IOU metrics use our G-Code renderer to measure translation quality.*

| Model | Training Data | | | IOU Metrics | | | |
|---|---|---|---|---|---|---|---|
| | *Files* | *Layers* | *Chunks* | **IOU@0.9** | **IOU@0.95** | **IOU@0.98** | **IOU@0.99** |
| GPT-2 Base | 0 | 0 | 0 | 67 | 61 | 17 | 4 |
| GPT-2$^{(1)}$ | 1 | 49 | 3933 | 95 | 88 | 71 | 27 |
| GPT-2$^{(5)}$ | 5 | 545 | 13371 | 96 | 91 | 74 | 30 |
| GPT-2$^{(25)}$ | 25 | 2298 | 51295 | 98 | 94 | 71 | 30 |

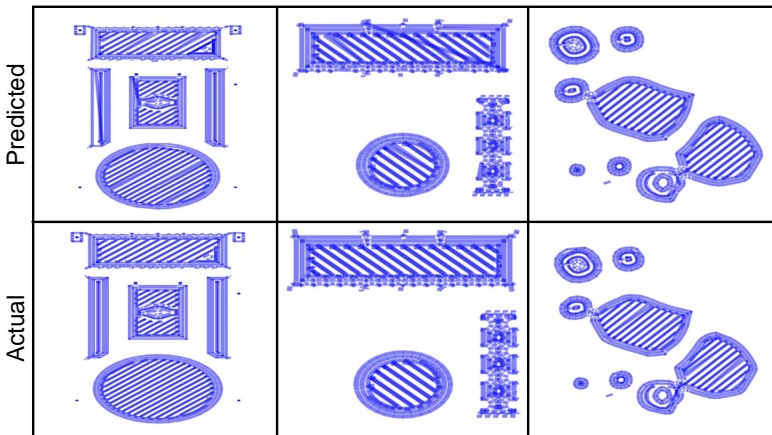

**Figure 5:** *Renderings of G-code layers predicted by a translation model finetuned on Slice-100K.*

Table 3 shows the results of performing finetuning on differently sized subsets of Slice-100K. We denote a GPT-2 model finetuned on $n$ shapes from our dataset as GPT-2$^{(n)}$. Figure 5 shows example renderings of shapes that have undergone flavor translation by our GPT-2$^5$ model. We also include an example G-code that has been translated in the Appendix (Figure 11). We find that even finetuning on minimal subsets of our dataset leads to significantly enhanced G-code translation abilities. Increasing the amount of training data beyond just five G-code shapes finetuning ceases to yield improvements. We attribute this to our preprocessing methods, which reduce the complex translation task to a simple local mapping, thereby reducing the amount of data needed for learning.

## 5 Conclusion

In this paper, we presented Slice-100K, the first large-scale, curated dataset of over 100,000 G-code files, along with their corresponding STL CAD files, renderings, and geometric properties. This dataset addresses a significant gap in the availability of comprehensive resources for additive manufacturing. We showcase our dataset explorer in Figure 6 and various modalities offered by Slice-100K in Figure 7. Our evaluation demonstrated the usefulness of Slice-100K in utilizing existing language models for tasks such as G-code debugging, geometric transformations, and comprehension. Additionally, we introduced a novel application of using Slice-100K for LLM-based G-code flavor translation, showcasing the potential of our dataset in advancing the field. We believe that Slice-100K will serve as a foundational resource for future innovations in manufacturing, paving the way for the development of domain-specific foundation models.

**Limitations:** Despite these advancements, Slice-100K has certain limitations. One major challenge is the difficulty in verifying the LVIS categories of the models. Additionally, all models in Slice-100K were sliced along the default Z-direction. This uniform slicing approach may limit the dataset's applicability for research into multi-directional slicing techniques and their impact on manufacturing outcomes. Furthermore, we primarily focus on extrusion-based 3D printing among a plethora of additive manufacturing techniques. Addressing these limitations in future versions of the dataset will be crucial for further enhancing its utility and broadening its application scope.

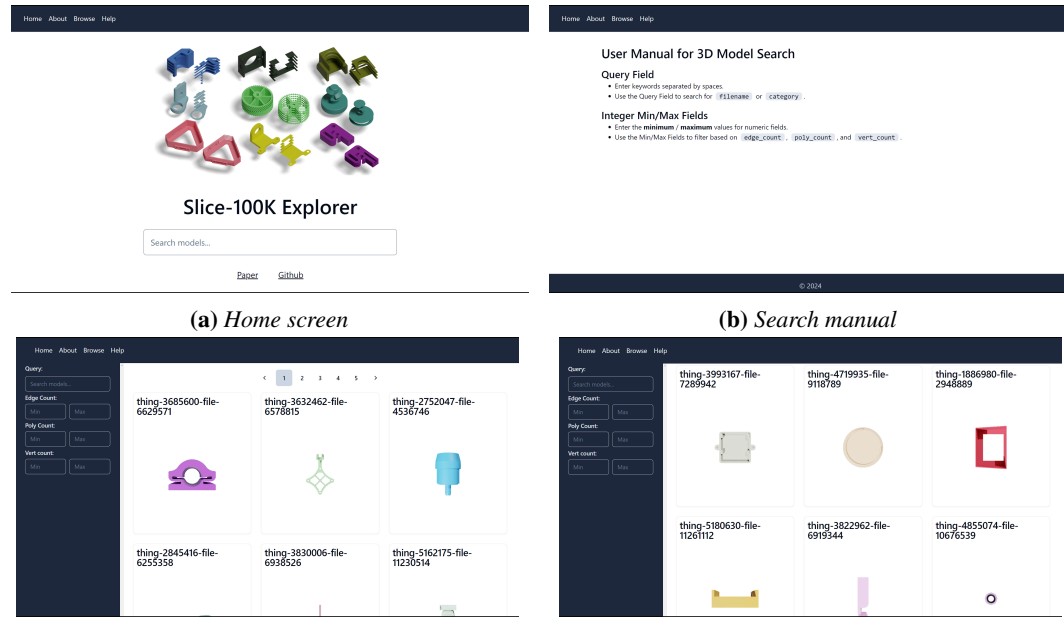

**(a)** *Home screen*

**(b)** *Search manual*

**(c)** *Model view*

**Figure 6:** *Dataset explorer screenshots: (a) Home screen, (b) Search manual, and (c) Model view.*

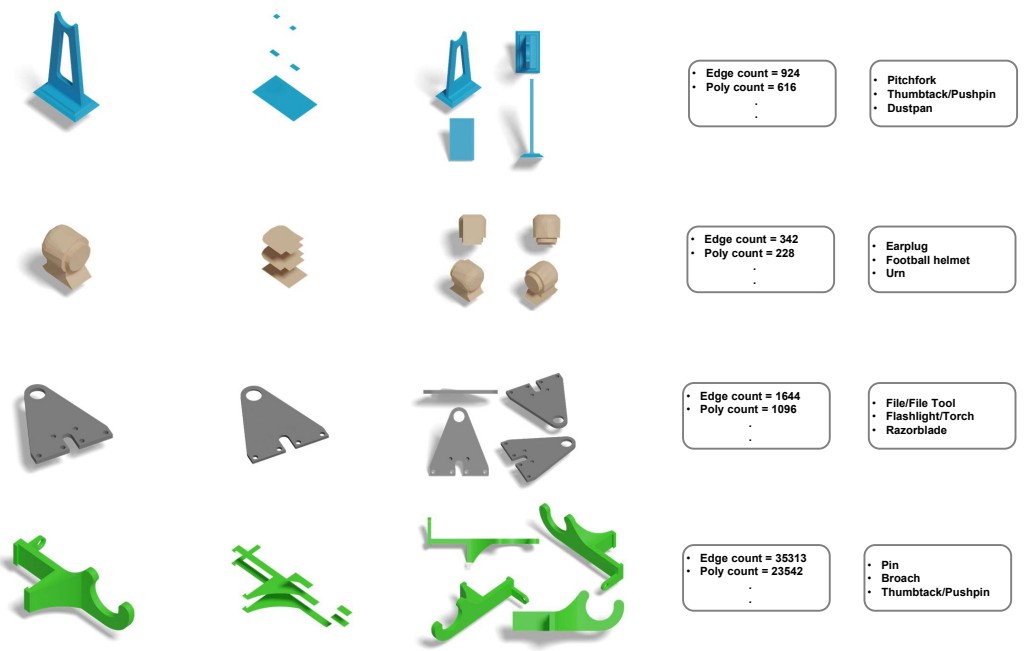

**Figure 7:** *Various modalities offered by Slice-100K. **First column** - CAD models, **Second column** - G-code, **Third column** - Renderings, and **Fourth and fifth columns** - Geometric properties and LVIS categories.*

## Acknowledgements

This work was supported by the National Science Foundation under grant CMMI-2347623/2347624. We would like to thank the NVIDIA Corporation for providing GPUs for academic research.

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

# A Appendix

---

**Algorithm 1** Contour Flipping

---

1: **procedure** CONTOURFLIP($LayerA, LayerB$)
2:     $ContoursA \leftarrow ContourSplit(LayerA)$
3:     $ContoursB \leftarrow ContourSplit(LayerB)$
4:     $Lookup \leftarrow HashMap()$              ▷ Create hash index of contours in Layer B
5:     **for** $c \leftarrow 1$ to **length**($ContoursB$) **do**
6:         **for** $i \leftarrow 1$ to **length**($ContoursB[c]$) **do**
7:             $line_i \leftarrow representation(ContoursB[c][i])$
8:             **if** $line_i \in Lookup$ and $Lookup[line_i] \neq i$ **then**
9:                 Delete $Lookup[line_i]$
10:             **else**
11:                 $Lookup[line_i] \leftarrow c$
12:             **end if**
13:         **end for**
14:     **end for**
15:     $Mapping \leftarrow HashMap()$              ▷ Find bijection between layers
16:     **for** $c_A \leftarrow 1$ to **length**($ContoursA$) **do**
17:         **for** $i \leftarrow 1$ to **length**($ContoursA[c_A]$) **do**
18:             $line_i \leftarrow representation(ContoursA[c_A][i])$
19:             **if** $line_i \in Lookup$ **then**
20:                 $c_B \leftarrow Lookup[line_i]$
21:                 $Mapping[c_B] \leftarrow c_A$
22:             **end if**
23:         **end for**
24:     **end for**
25:     $FlippedB \leftarrow Array[\textbf{length}(ContoursB[c_A])]$
26:     **for** $i \leftarrow 1$ to **length**($ContoursB$) **do**
27:         $FlippedB[Mapping[c_i]] \leftarrow ContoursB[i]$
28:     **end for**
29:     **return** $LayerA, FlippedB$
30: **end procedure**

---

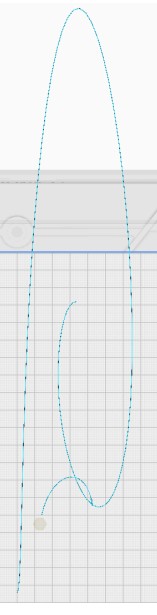

**Figure 8:** `GPT-3.5` *outlier case for scaling a cylinder.*

**Algorithm 2** Pair Creation

```
 1: procedure PAIR CREATION(LayerA, LayerB, maxLength)
 2:     start_i, start_j ← 0, 0
 3:     end_i ← maxLength
 4:     pairs ← List()
 5:     while start_i <= length(LayerA) do
 6:         end_j ← start_j + 1
 7:         found ← False
 8:         while ¬found and (end_j − start_j) ≤ maxLength do
 9:             if representation(LayerA[end_i]) = representation(LayerB[end_j]) then
10:                 found ← True
11:             end if
12:             end_j = end_j + 1
13:         end while
14:         if found then                                          ▷ Add matching pair of chunks to dataset
15:             chunk_a ← LayerA[start_i : end_i]
16:             chunk_b ← LayerB[start_j : end_j]
17:             pairs.append((chunk_a, chunk_b))
18:         else
19:             end_i = end_i − 1       ▷ Could not find a line matching line end_i, try a smaller chunk
20:         end if
21:     end while
22:     return LayerA, FlippedB
23: end procedure
```

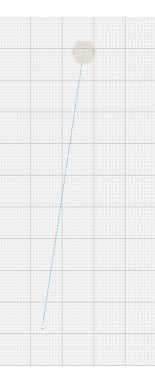

**Figure 9:** `Llama-2-70b` *outlier case for scaling a cylinder*

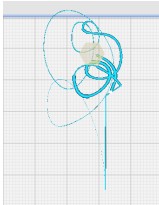

**Figure 10:** `Claude-2` *outlier case for Scaling S-shape.*

| Sailfish Input | Real Marlin | Predicted Marlin |
|---|---|---|
| G1 X57.724 Y65.24 E308.61973 | G1 X57.724 Y65.24 E33.07738 | G1 X57.724 Y65.24 E33.07738 |
| G1 X57.724 Y131.501 E311.34178 | G1 X57.724 Y131.501 E35.79943 | G1 X57.724 Y131.501 E35.79943 |
| G1 X57.793 Y131.939 E311.36 | G1 X57.793 Y131.939 E35.81765 | G1 X57.793 Y131.939 E35.81765 |
| G1 X58.031 Y132.411 E311.38172 | G1 X58.031 Y132.411 E35.83937 | G1 X58.031 Y132.411 E35.83937 |
| G1 X58.416 Y132.772 E311.4034 | G1 X58.416 Y132.772 E35.86105 | G1 X58.416 Y132.772 E35.86105 |
| G1 X58.902 Y132.981 E311.42513 | G1 X58.902 Y132.981 E35.88278 | G1 X58.902 Y132.981 E35.88278 |
| G1 X59.249 Y133.026 E311.4395 | G1 X59.249 Y133.026 E35.89715 | G1 X59.249 Y133.026 E35.89715 |
| G1 X140.751 Y133.026 E314.78766 | G1 X140.751 Y133.026 E39.24531 | G1 X140.751 Y133.026 E39.24531 |
| G1 X141.189 Y132.957 E314.80588 | G1 X141.189 Y132.957 E39.26353 | G1 X141.189 Y132.957 E39.26353 |
| G1 X141.661 Y132.719 E314.8276 | G1 X141.661 Y132.719 E39.28525 | G1 X141.661 Y132.719 E39.28525 |
| G1 X142.022 Y132.334 E314.84928 | G1 X142.022 Y132.334 E39.30693 | G1 X142.022 Y132.334 E39.30693 |
| G1 X142.231 Y131.848 E314.87101 | G1 X142.231 Y131.848 E39.32866 | G1 X142.231 Y131.848 E39.32866 |
| G1 X142.232 Y131.836 E314.8715 | G1 X142.232 Y131.836 E39.32915 | G1 X142.232 Y131.836 E39.32915 |
| G1 X141.921 Y131.736 F7800 | G1 X141.921 Y131.736 F7800 | G1 X141.921 Y131.736 F7800 |
| ;TYPE:External perimeter | ;TYPE:External perimeter | ;TYPE:External perimeter |
| G1 F1800 | G1 F1800 | G1 F1800 |
| G1 X141.947 Y131.501 E314.88121 | G1 X141.947 Y131.501 E39.33886 | G1 X141.947 Y131.501 E39.33886 |
| G1 X141.947 Y64.911 E317.61678 | G1 X141.947 Y64.911 E42.07443 | G1 X141.947 Y64.911 E42.07443 |
| G1 X145.553 Y64.911 E317.76492 | G1 X145.553 Y64.911 E42.22257 | G1 X145.553 Y64.911 E42.22257 |
| G1 X145.553 Y66.714 E317.83899 | G1 X145.553 Y66.714 E42.29664 | G1 X145.553 Y66.714 E42.29664 |
| G1 X145.553 Y134.858 E320.63839 | G1 X145.553 Y134.858 E45.09604 | G1 X145.553 Y134.858 E45.09604 |

**Figure 11:** *Example of our translation model converting Sailfish G-code to Marlin G-code*

