# Slice-100K: A Multimodal Dataset for Extrusion-based 3D Printing
# Supplementary Material

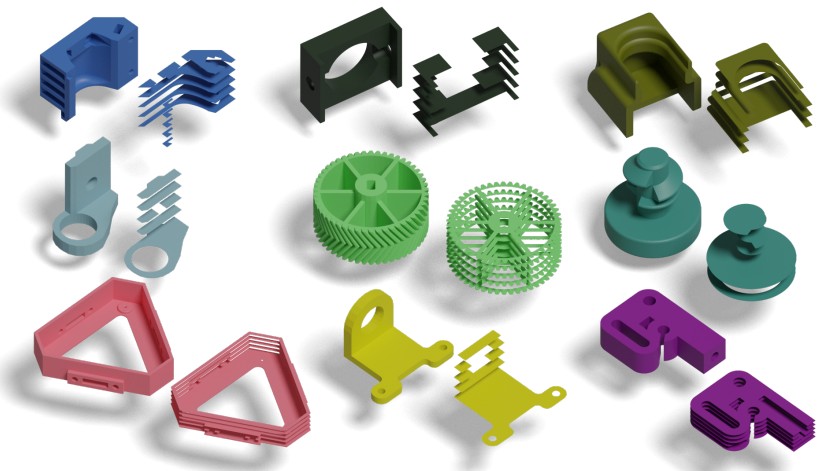

**Figure 1:** *The Slice-100K dataset consists of STL files and their G-code counterparts. Each pair here consists of STL (left) and its slices (right) for G-code.*

## 1 Overview

We provide additional details about the main paper and include the following in this document:

## 2 Societal Impacts

Slice-100K will provide a foundational platform that enables broader research at the intersection of manufacturing and artificial intelligence, particularly with the recent advances in multimodal large language models (LLMs). Furthermore, openly-available data is essential for the democratization of knowledge in the scientific community. Our dataset will facilitate a community-driven open-source AI ecosystem focused on utilizing multimodal manufacturing data to address challenges and increase

38th Conference on Neural Information Processing Systems (NeurIPS 2024) Track on Datasets and Benchmarks.

the feasibility of producing complex designs. We believe Slice-100K will benefit the economy and will be a net positive contribution.

However, we do foresee some potential negative societal impacts. A primary concern is that the dataset could enable advanced AI algorithms to generate G-code for dangerous objects. Safeguards have to be introduced to avoid such scenarios.

## 3 Slice-100K Analysis

We provide additional visualizations to understand the distribution of STL models in Slice-100K. The log-scale density plots of the number of vertices, polygons, and edges is shown in Figure 2, suggesting that a majority of the models possess at least $\sim 10^4$ vertices, polygons, and faces.

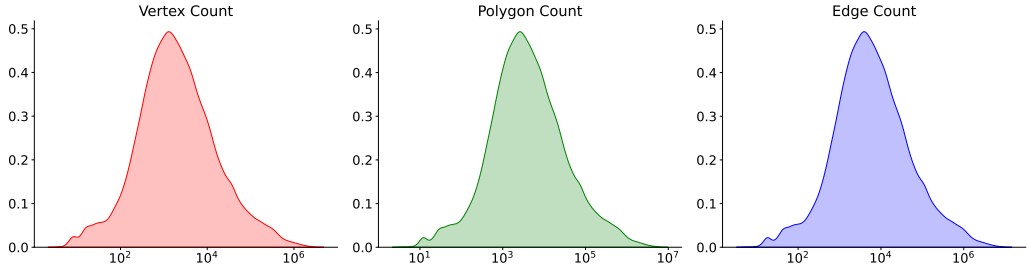

**Figure 2:** *Statistics of STL models present in Slice-100K. We show log-scale plots of number of vertices, number of polygons, and number of edges.*

We generated a word cloud (Figure 3) from the captions predicted as part of our LVIS category generation pipeline. We see *eraser, flashlight/torch* being highlighted more than others, suggesting the presence of models that are cuboidal or cylindrical in a general sense.

**Slicing:** We utilize Prusa's Slicer for generating G-code from STL files. Before slicing, we randomly select one of the four infill patterns shown in Figure 4. We introduce four infill patterns to encourage diversity among our G-code files as well as potentially mitigate bias during the finetuning process.

**Finetuning implementation:** For finetuning our translation model, we use a batch size of 32 with 8 gradient accumulation steps. Our learning rate is $1.4 \times 10^{-5}$. The model checkpoint to evaluate is chosen based on performance on our held-out set. We finete using 4 NVIDIA A100s. We include additional failure cases for G-code translation from Sailfish to Marlin on our finetuned model in Figure 5. Our finetuned model output is the same as the input since the model ends up ignoring Marlin-specific commands. We believe this might be due to a variety of factors, including tokenization not being able to address minute differences or the presence of contextual ambiguity.

## 4 License Information

The dataset is distributed under the CC-BY-4.0 license. We recommend users to double check the individual license of models gathered from Objaverse-XL and Thingi10K before proceeding with downstream tasks.

## 5 Dataset Accessibility and Long-Term Preservation Plan

The dataset is available for viewing on our project page. Our dataset is stored on Figshare and can be accessed here, and the scripts used for processing the data as well as our experiments can be accessed on Github. We will work with our institution to ensure the accessibility and long-term preservation of our dataset on Figshare.

## 6 Structured Metadata

We make use of an existing data repository to host our dataset. The structured metadata is automatically generated.

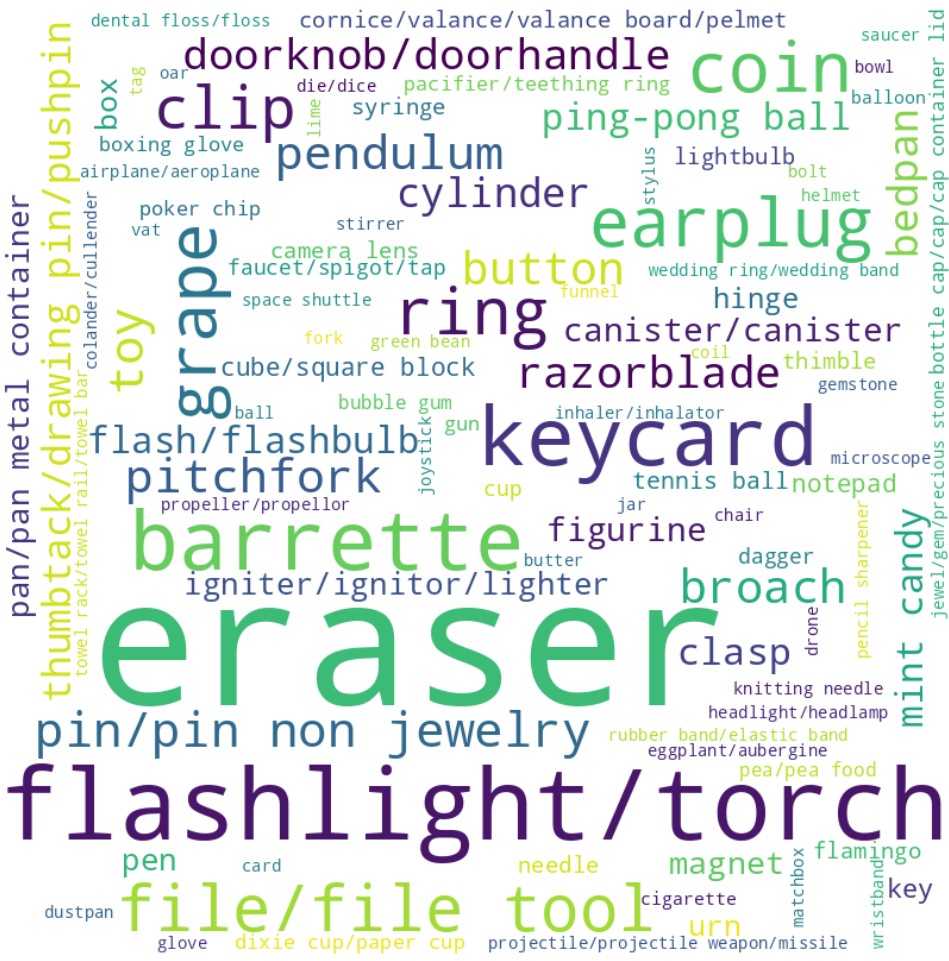

**Figure 3:** *Word cloud of categories the STL models in Slice-100K. These categories were generated as part of the text category generation pipeline.*

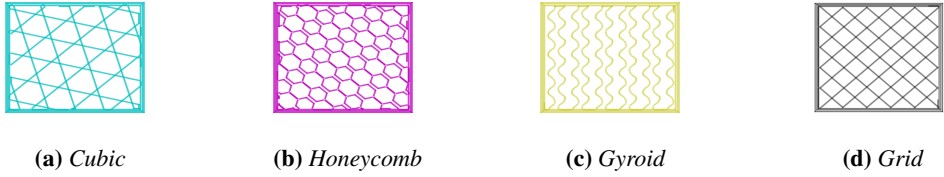

(a) *Cubic*     (b) *Honeycomb*     (c) *Gyroid*     (d) *Grid*

**Figure 4:** *Illustration of various infill patterns we utilized for slicing. During the slicing process, a random selection is made from the above shown patterns.*

## 7 Dataset Identifier

We have reserved a DOI and will provide upon acceptance, since providing it here will violate confidentiality.

## 8 Author Statement of Responsibility

The authors confirm they take all responsibility in case of violation of rights and confirm the license associated with Slice-100K.

| Sailfish | Predicted Marlin | True Marlin |
|---|---|---|
| ;HEIGHT:0.35
M126
G1 F3600
G1 X94.756 Y82.629 E1350.8806
G1 X94.749 Y82.47 E1350.88697
M126
;TYPE:External perimeter
;WIDTH:0.39375
;HEIGHT:0.3
G1 F1800
G1 X94.746 Y82.426 E1350.88878
G1 X94.693 Y82.39 E1350.89141
;WIDTH:0.36621
G1 X94.639 Y82.354 E1350.89385
;WIDTH:0.388659
G1 X94.347 Y82.343 E1350.90567
G1 X94.14 Y82.756 E1350.92435
G1 X94.026 Y82.067 E1350.9526
;WIDTH:0.357263
G1 X92.981 Y74.69 E1351.22477
;WIDTH:0.354691 | ;HEIGHT:0.35
M126
G1 F3600
G1 X94.756 Y82.629 E0.00545
G1 X94.749 Y82.47 E0.00637
M126
;TYPE:External perimeter
;WIDTH:0.39375
;HEIGHT:0.3
G1 F1800
G1 X94.746 Y82.426 E0.00181
G1 X94.693 Y82.39 E0.00263
;WIDTH:0.36621
G1 X94.639 Y82.354 E0.00244
;WIDTH:0.388659
G1 X94.347 Y82.343 E0.01182
G1 X94.14 Y82.756 E0.01868
G1 X94.026 Y82.067 E0.02825
;WIDTH:0.357263
G1 X92.981 Y74.69 E0.27217
;WIDTH:0.354691 | ;HEIGHT:0.35
M106 S255
G1 F3600
G1 X94.756 Y82.629 E3.60444
G1 X94.749 Y82.47 E3.61081
M106 S209.1
;TYPE:External perimeter
;WIDTH:0.39375
;HEIGHT:0.3
G1 F1800
G1 X94.746 Y82.426 E3.61262
G1 X94.693 Y82.39 E3.61525
;WIDTH:0.36621
G1 X94.639 Y82.354 E3.61769
;WIDTH:0.388659
G1 X94.347 Y82.343 E3.62951
G1 X94.14 Y82.756 E3.64819
G1 X94.026 Y82.067 E3.67644
;WIDTH:0.357263
G1 X92.981 Y74.69 E3.94861
;WIDTH:0.354691 |
| G1 F1800
G1 X108.054 Y126.557 E1348.89683
;WIDTH:0.399113
G1 X108.054 Y123.379 E1349.02951
G1 E1347.02951 F2400
G1 X108.054 Y123.379 F7800
G1 X108.155 Y126.996
G1 E1349.02951 F2400
;WIDTH:0.416819
G1 F900
G1 X108.063 Y126.983 E1349.03359
G1 X108.055 Y126.586 E1349.05104
G1 X108.202 Y126.732 E1349.06015
G1 X108.224 Y126.72 E1349.06125
;WIDTH:0.391194
G1 X108.247 Y126.708 E1349.06231
;WIDTH:0.365569
G1 X110.921 Y126.708 E1349.16276
G1 X110.921 Y126.858 E1349.16839
G1 X110.921 Y127.009 E1349.17406
G1 X108.247 Y127.009 E1349.27451 | G1 F1800
G1 X108.054 Y126.557 E0.00122
;WIDTH:0.399113
G1 X108.054 Y123.379 E0.13268
G1 E1347.02951 F2400
G1 X108.054 Y123.379 F7800
G1 X108.155 Y126.996
G1 E1349.02951 F2400
;WIDTH:0.416819
G1 F900
G1 X108.063 Y126.983 E0.00408
G1 X108.055 Y126.586 E0.01745
G1 X108.202 Y126.732 E0.00911
G1 X108.224 Y126.72 E0.0011
;WIDTH:0.391194
G1 X108.247 Y126.708 E0.00106
;WIDTH:0.365569
G1 X110.921 Y126.708 E0.10045
G1 X110.921 Y126.858 E0.00563
G1 X110.921 Y127.009 E0.00567
G1 X108.247 Y127.009 E0.10045 | G1 F1800
G1 X108.054 Y126.557 E2.00122
;WIDTH:0.399113
G1 X108.054 Y123.379 E2.1339
G1 E.1339 F2400
G92 E0
G1 X108.054 Y123.379 F7800
G1 X108.155 Y126.996
G1 E2 F2400
;WIDTH:0.416819
G1 F900
G1 X108.063 Y126.983 E2.00408
G1 X108.055 Y126.586 E2.02153
G1 X108.202 Y126.732 E2.03064
G1 X108.224 Y126.72 E2.03174
;WIDTH:0.391194
G1 X108.247 Y126.708 E2.0328
;WIDTH:0.365569
G1 X110.921 Y126.708 E2.13325
G1 X110.921 Y126.858 E2.13888
G1 X110.921 Y127.009 E2.14455
G1 X108.247 Y127.009 E2.245 |

**Figure 5:** *Failure cases for our translation model. The model's prediction is the same as its input and ignores Marlin-specific commands*

# 9  Datasheet

We provide a datasheet [1] for Slice-100K.

## 9.1  Motivation

- **For what purpose was the dataset created?**
  We created Slice-100K to provide a first-of-its-kind curated multimodal dataset for reference) of G-code, CAD models, and renderings to facilitate the application of VLMs for additive manufacturing. Slice-100K will encourage the research community to address new problems in design and manufacturing. Our dataset, built using models from Objaverse-XL and the Thingi10k dataset, encompasses a diverse range of 3D printable objects and provides a comprehensive resource for training a manufacturing domain-specific foundation model.

- **Who created the dataset (e.g., which team, research group) and on behalf of which entity (e.g., company, institution, organization)?**
  The authors listed on this paper. Anonymized for review.

- **Who funded the creation of the dataset?**
  Anonymized for review.

- **Any other comments?**
  The details will be de-anonymized upon acceptance.

## 9.2  Composition

- **What do the instances that comprise the dataset represent?**
  Each instance comprises of G-code, corresponding STL file, renderings of the STL along with metadata that includes the LVIS categories.

- **How many instances are there in total?**
  100,068.

- **Does the dataset contain all possible instances or is it a sample (not necessarily random) of instances from a larger set?**
  The dataset contains a subset of instances from Objaverse-XL's Thingiverse branch and the Thingi10K dataset.

- **What data does each instance consist of?**
  Each instance consists of a 3D printable part STL file, G-code files of the part, renders of the part, and metadata that includes LVIS categories.

- **Is there a label or target associated with each instance?**
  No. However, labels and target can be derived if a user is interested in performing language modeling related tasks using our dataset.

- **Is any information missing from individual instances?**
  No.

- **Are relationships between individual instances made explicit?**
  We treat each instance as independent. The LVIS categories are used to group them for viewing/indexing purposes on our website.

- **Are there recommended data splits (e.g., training, development/validation, testing)?**
  No. They could vary based on the downstream task.

- **Are there any errors, sources of noise, or redundancies in the dataset?**
  Not to our knowledge. However, if we discover any, we will fix them.

- **Is the dataset self-contained, or does it link to or otherwise rely on external resources (e.g., websites, tweets, other datasets)?**
  It is self-contained.

- **Does the dataset contain data that might be considered confidential?**
  It would be rare, however there is a possibility since we build our dataset using a subset of models gathered from Objaverse-XL and Thingi10K.

- Does the dataset contain data that, if viewed directly, might be offensive, insulting, threatening, or might otherwise cause anxiety?
  There is a rare possibility of finding data that might be considered offensive, threatening, or might cause anxiety. We will remove any such instances if we find them in the future.

- Does the dataset identify any subpopulations (e.g., by age, gender)?
  Not Applicable.

- Is it possible to identify individuals (i.e., one or more natural persons), either directly or indirectly (i.e., in combination with other data) from the dataset?
  There is rare possibility of a human-like model appearing in our dataset, but we believe it would be diffficult to identify individuals.

- Does the dataset contain data that might be considered sensitive in any way (e.g., data that reveals race or ethnic origins, sexual orientations, religious beliefs, political opinions or union memberships, or locations; financial or health data; biometric or genetic data; forms of government identification, such as social security numbers; criminal history)?
  There is a rare possibility that some models might have some cultural connotations. We will remove any such models that gets reported as being sensitive.

- Any other comments? No.

## 9.3 Collection

- How was the data associated with each instance acquired?
  The STL models were downloaded from Objaverse-XL's Thingiverse branch and from the Thingi10K dataset. G-code was generated by slicing the gathered STL files using Prusa's slicer. Renderings of STL files were generated using Blender, and captions and LVIS categories were generated using renderings of the STL files.

- What mechanisms or procedures were used to collect the data?
  Python scripts were used for data collection and curation. We have included all the scripts in the associated Git.

- If the dataset is a sample from a larger set, what was the sampling strategy?
  The dataset is filtered from Objaverse-XL's Thingiverse branch and Thingi10K. For the former, we only download files in the STL format and also check if files can be rendered using Blender. For the latter, we use `num components = 1`, `is manifold`, and `is oriented` for filtering. These were used to make sure that the parts are 3D printable.

- Who was involved in the data collection process and how were they compensated?
  The data curation process were fully performed by the authors of this paper.

- Over what timeframe was the data collected?
  Q4 of 2023, and Q1 & Q2 of 2024

- Were any ethical review processes conducted (e.g., by an institutional review board)?
  Not Applicable.

- Did you collect the data from the individuals in question directly, or obtain it via third parties or other sources (e.g., websites)?
  Data was collected from Objaverse-XL and Thingi10K. Both are publicly available resources.

- Were the individuals in question notified about the data collection?
  Not Applicable.

- Did the individuals in question consent to the collection and use of their data?
  The existing license of Objaverse-XL and Thingi10k allows for the use of the data.

- If consent was obtained, were the consenting individuals provided with a mechanism to revoke their consent in the future or for certain uses?
  Not Applicable.

- Has an analysis of the potential impact of the dataset and its use on data subjects (e.g., a data protection impact analysis) been conducted?
  No.

- Any other comments? No.

## 9.4 Processing/Cleaning/Labeling

- Was any preprocessing/cleaning/labeling of the data done (e.g., discretization or bucketing, tokenization, part-of-speech tagging, SIFT feature extraction, removal of instances, processing of missing values)?
  No preprocessing was done. However, post G-code generation and STL rendering, a portion of STL files were removed, owing to failure in the G-code generation process and rendering.

- Was the "raw" data saved in addition to the preprocessed/cleaned/labeled data (e.g., to support unanticipated future uses)?
  Yes.

- Is the software that was used to preprocess/clean/label the data available?
  Yes, we provide all code used to curate the data.

- Any other comments? No.

## 9.5 Uses

- Has the dataset been used for any tasks already?
  Yes. We showcase some applications in Section 4 of the main paper.

- Is there a repository that links to any or all papers or systems that use the dataset?
  No other research groups have used our dataset yet. We do have other publications that use the dataset, which we will refrain from adding here to maintain the confidentiality of the review process.

- What (other) tasks could the dataset be used for?
  Apart from G-code translation, we believe Slice-100K will serve as a foundation for future applications that train machine learning algorithms for manufacturing.

- Is there anything about the composition of the dataset or the way it was collected and preprocessed/cleaned/labeled that might impact future uses?
  We ask users to double check the license of the individual STL part files.

- Are there tasks for which the dataset should not be used?
  Not to our knowledge. However, the licensing information should always be double checked before use.

- Any other comments? No.

## 9.6 Distribution

- Will the dataset be distributed to third parties outside of the entity (e.g., company, institution, organization) on behalf of which the dataset was created?
  Yes. The dataset will be publicly available along with all the associated scripts.

- How will the dataset will be distributed (e.g., tarball on website, API, GitHub)?
  The dataset will be available for download from our project page. The associated scripts will be made available via our GitHub page.

- When will the dataset be distributed?
  The dataset will be made public after the paper review process.

- Will the dataset be distributed under a copyright or other intellectual property (IP) license, and/or under applicable terms of use (ToU)?
  The dataset is distributed under CC-BY-4.0 license. The individual parts are subject to the licenses that they are released under and a user needs to verify these licenses before any intended downstream task.

- Have any third parties imposed IP-based or other restrictions on the data associated with the instances?
  No.

- Do any export controls or other regulatory restrictions apply to the dataset or to individual instances?
  No.

- Any other comments? No.

## 9.7 Maintenance

- Who will be supporting/hosting/maintaining the dataset?
  The code for finetuning, rendering, slicing, and caption generation is on our publicly available GitHub repository. The dataset is currently available via an anonymous link and will be made available via our project page. There will be institutional support for hosting the dataset.

- How can the owner/curator/manager of the dataset be contacted (e.g., email address)?
  The corresponding author can be contacted via email.

- Is there an erratum?
  Not yet. However, if we discover any errors, we will update our project page with an erratum.

- Will the dataset be updated (e.g., to correct labeling errors, add new instances, delete instances)?
  Yes, if any errors are discovered, we will update the dataset.

- If the dataset relates to people, are there applicable limits on the retention of the data associated with the instances (e.g., were the individuals in question told that their data would be retained for a fixed period of time and then deleted)?
  The dataset does not relate to people. However, if anyone finds a specific part as sensitive, we will remove it on request.

- Will older versions of the dataset continue to be supported/hosted/maintained?
  Yes. All data will be version controlled, as long as they do not relate to sensitive data.

- If others want to extend/augment/build on/contribute to the dataset, is there a mechanism for them to do so?
  Yes, we provide scripts for slicing, rendering, and caption generation. We can be contacted via email and/or GitHub issues. We encourage the community to add to the dataset. We will work with interested parties to enable this addition.

## References

[1] Timnit Gebru, Jamie Morgenstern, Briana Vecchione, Jennifer Wortman Vaughan, Hanna Wallach, Hal Daumé Iii, and Kate Crawford. Datasheets for datasets. *Communications of the ACM*, 64 (12):86–92, 2021.