# OpenReview forum: "Slice-100K: A Multimodal Dataset for Extrusion-based 3D Printing"
_NeurIPS.cc/2024/Datasets_and_Benchmarks_Track — NeurIPS 2024 Track Datasets and Benchmarks Poster_

### Official Review · Reviewer_CqL5 · 2024-07-24
**SLICE-100K Review**

**Rating:** 7
**Confidence:** 4
**Correctness:** Yes
**Clarity:** Yes

**Review:**

Overall the paper is well-written and has provided details on data collection and experimental demonstrations. I believe SLICE-100K will be able to support later studies in digital manufacturing.

**Strengths:**

The paper and the dataset website are well-organized. The paper also provided details on data collection and the G-Code generation/translation process.

**Additional Feedback:**

The authors mentioned generating G-code is non-trivial and has many "flavors" in the introduction section. But later in Section 3.1, the generated G-code simply becomes of high quality and suitable for different 3D printers with PrusaSlicer. I'm wondering how the "high quality" and the generalizability are ensured. Is there any human labor required to check the G-code rendering results for this process?

**Documentation:**

Yes

**Opportunities For Improvement:**

A minor question that still confused me after reading the paper is how many flavors are included in the proposed dataset. Does each data point include multiple flavored G-Codes (or say at least Sailfish and Marlin)?  I might miss this information somewhere; please point me out if it has been mentioned in the paper.

Also, the LVIS quality was a concern for me. It seems the category tags are not very accurate on the dataset website. The author has also mentioned this in the Limitation section. I'm wondering if there are any potential solutions to improve the quality of the LVIS categories.

**Relation To Prior Work:**

Yes

**Summary And Contributions:**

This paper presents SLICE-100K, a multimodal dataset for 3D printing. Each data point contains the 3D stl model, rendering images, some geometry properties, category tags, and most importantly, the G-code indicating the 3D printer instructions. The experiments show that the existing chat-based LLMs cannot handle the G-Code geometric transformations flawlessly or acceptably. The LLMs can also benefit from the proposed dataset to perform G-code flavor translation tasks.

---

> ### Author Rebuttal · Authors · 2024-08-17
>
> **Opportunities For Improvement**
>
>
> > A minor question that still confused me after reading the paper is how many flavors are included in the proposed dataset. Does each data point include multiple flavored G-Codes (or say at least Sailfish and Marlin)? I might miss this information somewhere; please point me out if it has been mentioned in the paper.
>
> We thank the reviewer for their question. For each STL file we slice, we generate two flavors of G-code - Marlin and Sailfish. We mention this in line 192 under Section 3.2. Additionally, we will explicitly mention this fact in Section 3.1 under "G-code Generation."
>
> > Also, the LVIS quality was a concern for me. It seems the category tags are not very accurate on the dataset website. The author has also mentioned this in the Limitation section. I'm wondering if there are any potential solutions to improve the quality of the LVIS categories.
>
> We appreciate the reviewer's concern. As mentioned above, we agree that the LVIS category tags are not very accurate. One way to address the accuracy aspect would be to utilize a widely available large language model's API to generate text descriptions and extract descriptive words to serve as categories.
>
> **Additional Feedback**
>
> > The authors mentioned generating G-code is non-trivial and has many "flavors" in the introduction section. But later in Section 3.1, the generated G-code simply becomes of high quality and suitable for different 3D printers with PrusaSlicer. I'm wondering how the "high quality" and the generalizability are ensured. Is there any human labor required to check the G-code rendering results for this process?
>
> We thank the reviewer for their question. We would like to clarify that the "quality" of G-code depends on the specific use case for manufacturing. Ideally, a good quality G-code should be enable a high fidelity 3D print of the target CAD model. Additionally, most slicing software packages provide multiple preset configurations, each addressing a different aspect (such as speed, accuracy, and print time among many). Thus by "high-quality," we refer to the ability to customize slicing parameters to achieve the highest quality G-code for extrusion based 3D printing in our case. With respect to Slice-100K, for each model that we slice, we utilize the default slicing parameters (with the exception of infill pattern that is chosen randomly, thus encouraging generalizability). Before building the dataset, a small subset of combinations of infill patterns were manually checked, following which we proceeded with the full-scale dataset generation.

---

> > ### Comment · Reviewer_CqL5 · 2024-08-27
> >
> > Thanks for the author's explanation.

---

### Official Review · Reviewer_9nHc · 2024-07-24
**a large multimodal dataset for extrusion-based 3D printing**

**Rating:** 7
**Confidence:** 3
**Correctness:** Yes.
**Clarity:** Yes.

**Review:**

Pros:
* comprehensive and detailed dataset including G-code, CAD models, renderings, and metadata.
* robust methodology ensuring reproducibility and detailed documentation.
* demonstrates novel applications and potential for advancing 3D printing research.

Cons:
* limited diversity due to reliance on existing datasets.
* uniform slicing direction may not fully represent practical slicing techniques.
* specific focus on extrusion-based 3D printing may limit broader applicability.
* difficulty in verifying the LVIS categories for the models.

**Strengths:**

Please see above.

**Additional Feedback:**

Please see above.

**Documentation:**

Yes.

**Ethics:**

No.

**Limitations:**

Yes.

**Opportunities For Improvement:**

When browsing the dataset, some model names are not descriptive, such as 'thing-xxx.' The authors may consider assigning more meaningful and descriptive names to the models, which could enhance the querying and browsing experience and efficiency of the dataset.

**Relation To Prior Work:**

Yes.

**Summary And Contributions:**

This paper presents SLICE-100K, a first-of-its-kind dataset of over 100,000 G-code files, along with their tessellated CAD model, LVIS categories, geometric properties, and renderings. The dataset is built from triangle meshes derived from Objaverse-XL and Thingi10K datasets. GPT-2 is fined tuned on a subset of the dataset for G-code translation from a legacy G-code format to a more modern, widely used format.

---

> ### Author Rebuttal · Authors · 2024-08-17
>
> **Opportunities For Improvement**
>
> >When browsing the dataset, some model names are not descriptive, such as 'thing-xxx.' The authors may consider assigning more meaningful and descriptive names to the models, which could enhance the querying and browsing experience and efficiency of the dataset.
>
> We thank the reviewer for their feedback. Over time, we will update the model names to be more descriptive, allowing for a better overall interaction experience. We will also update the search process to incorporate categories as search terms, thus streamlining category-specific model retrieval from our proposed dataset.

---

### Official Review · Reviewer_YCzd · 2024-07-25
**This work proposes an automatic method to generate G-code for CAD models and introduces a new dataset with G-code features by combining two existing 3D CAD datasets.**

**Rating:** 6
**Confidence:** 3
**Correctness:** yes
**Clarity:** yes

**Review:**

Overall, I find the task interesting and the writing clear, with the method section being particularly easy to follow. However, the evaluation part lacks clarity. I am curious about how the quality of the generated LVIS and G-code is assessed. Specifically, it would be beneficial to understand the metrics or criteria used to evaluate the performance and accuracy of the generated data. Including more comparisons with ground truth data would greatly enhance the evaluation. This additional information would provide a clearer understanding of the effectiveness and reliability of the proposed method. Additionally, it would help in assessing how well the generated LVIS and G-code match up to actual data, offering a more robust validation of the approach.

**Strengths:**

The proposed dataset is vital for 3D CAD, particularly concerning the G-code. The authors suggest a method to automatically generate G-code for existing 3D CAD datasets.

**Additional Feedback:**

NA

**Documentation:**

yes

**Ethics:**

yes

**Opportunities For Improvement:**

Have more visualization and numerical metrics to validate the quality of G-code and LVIS. For example, attach more multi-view of 3D visualization of the reconstructed CAD 3D model by the G-code and the ground-truth CAD 3D model, Similly, it would be great if the authors could have more numeric comparison between the reconstructed CAD 3D model and the ground truth.

**Relation To Prior Work:**

yes

**Summary And Contributions:**

Proposing a foundational dataset for 3D CAD models is crucial as it significantly aids the design process. The authors emphasize an intriguing attribute of 3D CAD models: using G-code to bridge traditional 3D CAD models with generative processes.

---

> ### Author Rebuttal · Authors · 2024-08-17
>
> > Overall, I find the task interesting and the writing clear, with the method section being particularly easy to follow. However, the evaluation part lacks clarity. I am curious about how the quality of the generated LVIS and G-code is assessed. Specifically, it would be beneficial to understand the metrics or criteria used to evaluate the performance and accuracy of the generated data. Including more comparisons with ground truth data would greatly enhance the evaluation. This additional information would provide a clearer understanding of the effectiveness and reliability of the proposed method. Additionally, it would help in assessing how well the generated LVIS and G-code match up to actual data, offering a more robust validation of the approach
> > Have more visualization and numerical metrics to validate the quality of G-code and LVIS. For example, attach more multi-view of 3D visualization of the reconstructed CAD 3D model by the G-code and the ground-truth CAD 3D model, Similly, it would be great if the authors could have more numeric comparison between the reconstructed CAD 3D model and the ground truth.
>
>
> We thank the reviewer for their suggestions. Validating the quality of LVIS is indeed a major challenge in itself, and we acknowledge this in our limitations paragraph on the last page of the main paper. We utilize LVIS because (to the best of our knowledge) "ground-truth" categories for objects from Objaverse-XL have not been made public. For models from Thingi10K, we could in theory utilize the name tags, however, these name tags are not sufficient enough to provide a comprehensive evaluation of our LVIS categories. However, should Objaverse-XL categories become available in the future, we will integrate those categories and refine our dataset.
>
> Regarding the slicing process, we acknowledge that it is generally deterministic for a given set of slicing parameters. Specifically, for a known 3D geometry and a fixed set of slicing parameters, the process will consistently generate the same toolpath, assuming no randomization of the starting and ending points of each G-code layer. As a result, while we understand the importance of validation, we believe that computing traditional validation metrics for this aspect may not be necessary, as the deterministic nature of the slicing process ensures consistent outputs.

---

### Official Review · Reviewer_YQbc · 2024-07-29
**A Large-Scale Multimodal Dataset for 3D Printing**

**Rating:** 6
**Confidence:** 3

**Review:**

SLICE-100K is a novel contribution, addressing a significant gap in the field of AI for manufacturing. The combination of G-code, CAD models, and associated metadata in a large-scale dataset is unique.

This work could have substantial impact on the field of AI-assisted manufacturing and 3D printing. It provides a valuable resource for researchers and practitioners working on advanced manufacturing techniques.

Pros:

Large-scale, multimodal dataset combining G-code, CAD models, and metadata.

Practical demonstrations of dataset utility through LLM evaluations and fine-tuning.

Introduction of new evaluation metrics for G-code fidelity.

Potential to accelerate AI research in manufacturing.

Cons:

Limited to uniform Z-axis slicing, which may not represent all 3D printing scenarios.

Challenges in verifying LVIS categories for all models.

Experiments focused on a narrow range of tasks (scaling and flavor translation).

Possible biases in the dataset due to its sources (Thingiverse and Thingi10K).

**Strengths:**

Introduces the first large-scale, curated dataset of G-code files paired with CAD models and metadata for 3D printing research.

Addresses a critical gap in resources for AI applications in digital manufacturing.

Provides a foundation for developing more advanced AI models for 3D printing and manufacturing processes.

Enables interdisciplinary research at the intersection of AI, computer vision, and manufacturing.

Supports research in areas such as 3D object understanding, generative design, and automated manufacturing optimization.

**Additional Feedback:**

Consider including additional tasks beyond scaling and flavor translation to demonstrate the dataset's versatility.

How might SLICE-100K be used to develop AI systems that can optimize 3D printing processes for factors like print time, material usage, or structural integrity?

**Clarity:**

The paper is generally well-written.

Complex concepts like G-code processing and dataset creation are explained in a comprehensible manner.

The data collection process, G-code generation, and preprocessing methods are described in detail.

The paper would benefit from a more detailed discussion of potential future directions and improvements.

**Correctness:**

The SLICE-100K dataset is constructed in a sound manner, with clear methodologies for data collection, processing, and curation.

The evaluation methods and experiment design are appropriate and performed correctly.

**Documentation:**

The paper provides a clear explanation of the data sources (Objaverse-XL and Thingi10K) and the selection criteria.

The process of G-code generation using PrusaSlicer is well-described.

The dataset composition is clearly presented, including the number of objects from each source.

The paper describes the inclusion of renderings, LVIS categories, and geometric properties.

The paper describes some potential applications of the dataset, particularly in G-code manipulation and translation.

The paper provides detailed algorithms for contour flipping and pair creation, which supports reproducibility.

**Ethics:**

No concern.

**Limitations:**

The authors have partially addressed some limitations of their work.

There is no discussion of potential biases in the dataset due to its sources (Thingiverse and Thingi10K).

The paper doesn't address limitations in the diversity of 3D printing scenarios represented.

**Opportunities For Improvement:**

The dataset is limited to extrusion-based 3D printing, not covering other additive manufacturing techniques.

The evaluation of LLMs is limited to a small set of tasks (scaling and flavor translation), which may not fully demonstrate the dataset's potential.

The dataset is limited to uniform Z-axis slicing, which may not represent all 3D printing scenarios or more advanced slicing techniques.

**Relation To Prior Work:**

Table 1 provides a clear comparison of SLICE-100K with other 3D multimodal datasets, highlighting its unique inclusion of G-code files.

The authors state that SLICE-100K is a "first-of-its-kind dataset of over 100,000 G-code files" along with associated data, clearly positioning it as a novel contribution.

**Summary And Contributions:**

The paper introduces SLICE-100K, a novel dataset designed to advance AI applications in additive manufacturing and 3D printing.

Key contributions include: A curated collection of over 100,000 G-code files paired with corresponding STL CAD models, renderings, LVIS categories, and metadata. Evaluation of existing large language models (LLMs) on G-code geometric transformation tasks.

The authors position SLICE-100K as a foundational resource for developing AI models in digital manufacturing, addressing the lack of comprehensive datasets in this domain. They demonstrate its potential through experiments on G-code manipulation and translation tasks, showing improvements over baseline models even with limited training data.

---

> ### Author Rebuttal · Authors · 2024-08-17
>
> **Limitations**
>
> >The authors have partially addressed some limitations of their work.
>
> As other reviewers have also pointed out, a main limitation of our dataset is that the models are sliced in the default (Z-) direction. In a future update we will expand the dataset with the addition of random slicing directions as well as varying levels of adaptive slicing schemes. In addition, we curently are restricted to only 2 flavors of G-code. We plan to extend this to more flavors.
>
> >There is no discussion of potential biases in the dataset due to its sources (Thingiverse and Thingi10K).
>
> We thank the reviewer for this question. The main bias in the dataset is that it is likely limited to small-size CAD objects. However, we feel this is suitable for our use case since we are interested in a dataset of manufacturable parts.
>
> >The paper doesn't address limitations in the diversity of 3D printing scenarios represented.
>
> We agree with the reviwer that we have mainly focused on extrusion-based 3D printing (also primarily plastics). However, the underlying G-code can be extended to other 3D printing processes as well as for subtractive manufacturing. We will include a note in our revised paper.
>
>
> > Consider including additional tasks beyond scaling and flavor translation to demonstrate the dataset's versatility.
>
> We thank the reviewer for their suggestion. We certainly hope that the community can come up with novel use case for our dataset. These can include more complex tasks such as object identification, shape matching, and shape manipulation. This will be especially useful in a CAD and manufacturing setting where the user can quickly mine a queryable database and synthesize brand new versions or views.
>
> > How might SLICE-100K be used to develop AI systems that can optimize 3D printing processes for factors like print time, material usage, or structural integrity?
>
>
> G-code files are an alternate representation of the original geometry that is amenable to direct manufacturing. This representation can be leveraged to facilitate future studies involving print time optimization, material usage, or structural integrity. We envision that Slice-100K will grow over time and enable large-scale annotation and modernization of legacy G-code files. Additionally, it holds the potential for building AI models capable of matching and parsing similar parts/shapes to predict shape-class-specific properties such as print time, material usage, and structural integrity. A promising application lies in training a conversational AI model on Slice-100K, designed to provide real-time insights into process-specific parameters such as manufacturability and effectively acting as an AI sentry for manufacturing processes.

---

> > ### Comment · Reviewer_YQbc · 2024-08-24
> >
> > I acknowledge the rebuttal.

---

### Author Rebuttal · Authors · 2024-08-17

We thank the reviewers for their thoughtful feedback and their efforts. It is encouraging to hear positive feedback about our dataset. We have responded to each reviewer individually, addressing their questions and comments.

In addition, we have conducted GPT-2 fine-tuning on multiple shapes to provide a better understanding of the accuracy of our finetuning task. We found rough trends--e.g no fine tuning is worse than fine tuning by about 4 points of perplexity--although more training is needed to discover rigorous scaling laws.

---

> ### Author Response · Authors · 2024-08-27
> **Paper Discussion**
>
> We thank all the reviewers again for the detailed feedback. We would be happy to answer any follow-up questions before the discussion period ends on August 31.

---

### Decision · Program_Chairs · 2024-09-26

**Decision:**

Accept (Poster)

**Comment:**

This paper proposes Slice-100K, a dataset of 100K 3D models for 3D printing. For each model, the G-Code, STL, renderings, and LVIS category is provided. The 3D models are collected from Thingiverse (selected from models in Objaverse-XL  and Thingi10K).  Experiments are conducted on the dataset to investigate the ability of LLMs to generate and transform G-Codes.

All four reviewers were positive and recommended the paper for acceptance, noting that the paper was well-written, with clear description of the data processing.  Concerns with the dataset includes: limited to uniform slicing along the Z-axis (which only accommodates certain 3D printing techniques, and limits the applicability of the dataset for a broader set of slicing techniques for 3D printing), noise in the LVIS categories, and potential biases in the dataset.

As this is one of the first large scale datasets for 3D printing, the AC agrees that this work can be of value to the community.